# An Improved Optimization Model to Predict the MOR of Glulam Prepared by UF-Oxidized Starch Adhesive: A Hybrid Artificial Neural Network-Modified Genetic Algorithm Optimization Approach

**DOI:** 10.3390/ma15249074

**Published:** 2022-12-19

**Authors:** Morteza Nazerian, Jalal Karimi, Hossin Jalali Torshizi, Antonios N. Papadopoulos, Sepideh Hamedi, Elham Vatankhah

**Affiliations:** 1Department of Bio Systems, Faculty of New Technologies and Aerospace Engineering, Shahid Beheshti University, Tehran 1983969411, Iran; 2Department of Bio Refinery, Faculty of New Technologies and Aerospace Engineering, Shahid Beheshti University, Tehran 1983969411, Iran; 3Laboratory of Wood Chemistry and Technology, Department of Forestry and Natural Environment, International Hellenic University, GR-661 00 Drama, Greece

**Keywords:** glulam, UF-modified starch adhesive, ANN, genetic algorithm, optimization

## Abstract

The purpose of the present article is to study the bending strength of glulam prepared by plane tree (Platanus Orientalis-L) wood layers adhered by UF resin with different formaldehyde to urea molar ratios containing the modified starch adhesive with different NaOCl concentrations. Artificial neural network (ANN) as a modern tool was used to predict this response, too. The multilayer perceptron (MLP) models were used to predict the modulus of rapture (MOR) and the statistics, including the determination coefficient (R^2^), root mean square error (RMSE), and mean absolute percentage error (MAPE) were used to validate the prediction. Combining the ANN and the genetic algorithm by using the multiple objective and nonlinear constraint functions, the optimum point was determined based on the experimental and estimated data, respectively. The characterization analysis, performed by FTIR and XRD, was used to describe the effect of the inputs on the output. The results indicated that the statistics obtained show excellent MOR predictions by the feed-forward neural network using Levenberg–Marquardt algorithms. The comparison of the optimal output of the actual values obtained by the genetic algorithm resulting from the multi-objective function and the optimal output of the values estimated by the nonlinear constraint function indicates a minimum difference between both functions.

## 1. Introduction

Starch is a polysaccharide found in many plants, such as cereals, and is very important in industrial consumptions due to its cheapness, biodegradability, renewability, and safety. However, when applied as an adhesive, it has constraints, such as low shear and thermal resistance, thermal decomposition, and weak stability in solidification-melting. Hence, it must be treated, which can be mainly based on three main hydrothermal, chemical, and enzymic methods to improve its quality of application. 

The modified starch can be a suitable substitute for synthetic resins, such as UF from which formaldehyde gas can be emitted. Polycondensation of formaldehyde-based resins is the main method of making these resins due to their high reactivity, chemical adaptability, and economic competition [1]. The reaction between urea and formaldehyde is divided into three stages: 1—the alkaline formation of mono-, di-, and tri-methylol urea; 2—the acidic compression of methylols and destruction of some methylene-ether bonds to form methylene bridges [2]; 3—neutralizing pH and adding the final urea to adjust the F/U molar ratio. When adding the modified starch to any of the stages above, the formaldehyde molar ratio decreased due to the chemical change of the starch chain. 

When applying modified starch as an adhesive to produce wood-based composites, the process of the chemical treatment and oxidation of the starch polymer chains is very important. Modified starch is usually prepared in the reaction with oxidizing agents such as H_2_O_2_ or NaOCl under certain temporal, temperature, and pH conditions [3]. These oxidizing agents convert some OH-starch groups in C-2, C-3, or C-6 positions to carbonyl (−CO) or carboxyl (−COOH) groups [3]. During this process, NaOCl oxidizes hydroxyl groups to carbonyl and converts them to carboxyl groups afterwards [4]. Oxidation also modifies the molecular structure through polymerization [3,5]. A higher bonding strength and more water resistance can be obtained by combining starch with other synthetic resins, such as UF [6].

Due to the decrease in the crystallinity index and lower thermal stability, adding starch could improve the properties of wood panels adhered by the UF resin. Adding starch also results in a uniform distribution of the adhesive in the wood panels [7]. Luo et al. (2019) showed that dialdehyde starch is involved in the aminoplast resin synthesis by in situ polymerization between aldehyde groups in starch and amino group and hydroxymethyl in resin [8]. During this process, polymerization resulted in the penetration of the soft component of starch into the cross-linked network of rigid aminoplast resins and the formation of a microphase separated structure so that the wet shear strength improved in the plywood due to the increase in the cross-linking density, the curing acceleration of aminoplast resin, and the microphase separated structure. The diagnostic analyses showed that the wood composites made by the UF-modified starch adhesive had a lower intensity in the peaks of OH groups in the FTIR analysis due to the starch chemical modification, showing the increase in cross-linking density due to the decrease in the hydroxyl groups after the condensation reaction and formation of a 3D network in the curing process [9,10].

The connections made between two substrates by an adhesive layer are used normally in many consumptions. The efficiency of the connections from these adhesives is evaluated normally by common traditional experimental, analytical, or numerical methods. Due to the absence of valid conventional methods to present a certain phenomenon correctly, machine learning methods can be used to release the data-driven models [11,12]. In recent studies, artificial intelligent methods were used to replace the traditional methods to analyze the glue line bonding strength of the wood-based composites to make the predictive models [11,13]. These methods can help decrease the large number of experiments to achieve the optimal solution and decrease the cost. The artificial neural network (ANN) is a famous artificial intelligence method that can be used to model the complex relationship between the variables [11]. ANNs have become increasingly popular due to their ability to solve complex problems in different contexts. This technique can draw the complex relation between the variables related to each other without any presumption [14]. ANNs are suitable in cases with numerous variables and complex relationships whose definition is difficult with mathematical equations. When ANNs are trained correctly, they can quickly conclude from a data input and consequently offer acceptable solutions on the problem. It has made the ANN method superior to many modeling approaches, such as the analytical and numerical methods. The most attractive advantage of ANN is the capability of modeling multi-dimensional nonlinear and complex formulae without considering the relationship behavior [15,16]. These methods support the understanding of very uncertain designs helping predict their future outcomes by producing a stable mechanism in materials science research so that the modeling of processes, optimization, and regression can be used in different sections, including the materials structure [17,18]. 

Due to these advantages, ANNs have been used in different fields of wood science. Using the experimental test data obtained by the studies on many properties of wood and wood products, such as the veneer defects classification [19], the drying process [20], identification [21], and physical properties [22] when applying different independent factors, it was shown that, using different machine learning methods, the response being examined can be modeled successfully and an efficient method can be presented to identify, predict, and optimize different features of wood and wood-based composites. 

However, using the ANN methods has some disadvantages. During the modeling by ANN and its accuracy, two factors are very important: weight connecting the structure of ANN and modified quantities. This weight can have negative, positive, or zero values. With respect to the type of used ANN, these values, which are produced stochastically, may be real or an integer number [23]. The estimation of these values is difficult, basically. The wrong selection of weight may cause a drop into the local minimum and decrease the probability of detecting the optimum point. Additionally, it can decrease the convergence velocity of methods [23]. It needs to eliminate these limitations. Hence, it is useful to use GA in combination with ANN in order to reproduce network connection weights of the ANN optimization process. GA is a global iterative optimization method in which, by simulating the evolution of organisms, such as the selection and elimination colony that is selected and mutated, repeatedly [24]. The survival of the best and elimination of the worst is one of the most important evaluation and mutation laws, as well as an adaptive estimation of every individual in which the better colony is produced, gradually. The best individuals are optimized in the colony by global and parallel techniques [25]. Due to the validity of the solution, GA has strengthened the ability of global searching. GA is able to provide fast and enough solutions so that GA can be attractive for use in solving optimization problems. GA can omit the limitation of ANN. Hence, this technique in which ANN and GA are integrated is called GANN [26].

In the present research, the effect of the F to U molar ratio at three levels (MR; 1.5:1, 2:1 and 2.5:1), the weight ratio of the modified starch to UF resin at three levels (WR; 10, 20 and 30%), and the NaOCl solution concentration percent, as the starch treater to produce the starch adhesive used to make the UF-MS adhesive at three levels (AC; 10, 15 and 20%), as the independent input variables was evaluated on the bending strength of the three-layer glulam produced during the loading perpendicular on the glue line as the dependent output variable using the ANN technique. Using the genetic algorithm (GA) coupled with ANN and the multi-objective and nonlinear constraint functions, the optimum limit of the application of the independent variables to achieve the highest MOR was determined.

## 2. Materials and Methods

### 2.1. Chemical Modification of Starch

First, 100 gr corn starch powder was loaded in a flask containing 200 mL distilled water. While the mixture was mixed on a magnetic stirrer-heater, the NaOCl solution with 10%, 15%, or 20% concentration (7, 10, or 13 g) was added to the mixture, according to the test design used, and it was mixed for 30 min at the temperature 30 °C (with pH = 9.5). After being mixed well, the mixture was neutralized using sulfuric acid 20% and was put on the stirrer-heater for 10 min at the temperature 30 °C. After being put in the falcons, the resulting solution was centrifuged for 20 min at 2000 rpm to separate its water. Then, the starch deposited in the falcons was washed by distilled water for five times. To remove more moisture from the mixture, a Buchner funnel equipped with a vacuum pump was used and the obtained starch was dried on the filter paper at room temperature. This process was repeated with the different sodium hypochlorite percentages used in the chemical treatment of starch.

### 2.2. Synthesis of Starch Adhesive

The UF-MS adhesive synthesis was in two stages. According to the test design used, 42 g urea consumed at the first stage together with 120 cc, 150 cc, or 200 cc formalin (equivalent to 1.5 mol, 2 mol, or 2.5 mol formaldehyde, respectively) and 18 g, 24 g, or 30 g starch consumed to be added in the second stage were weighed. After installing the heater and setting the temperature in the oil bath and stabilizing the three-necked flask equipped with the condenser, thermometer, and pH-meter (blocked by cork) in the bath, formalin and the first part of urea were added to the flask while the mixture was mixed constantly by the magnetic stirrer. The mixture’s pH was increased to 8–8.5 by adding few drops of NaOH 0.5 mol with the concentration 20% and it was mixed for 15 min at the temperature 90–95 °C. To decrease the time of the reaction more, the mixture’s pH was decreased to 4–4.5 by adding sulfuric acid 1% (0.5–3 mL). The approximate period was 15 min at the temperature 90–95 °C. This time was enough to achieve a homogeneous single-phase mixture. After removing the condenser from the flask and opening two other necks, the extra water evaporated at the temperature 90–95 °C for 10–20 min. The formed acidic medium was neutralized by adding a few drops of NaOH. Decreasing the reaction medium to 50–60 °C, the starch which replaced urea at the second stage was added to the mixture at the three levels according to the test design. At this temperature, the mixture was mixed for 30 min until the starch dissolved completely. After cooling at room temperature, the formaldehyde urea adhesive was prepared.

### 2.3. Making Glulam

After cutting the plane tree (*Platanus Orientalis*-L) and slicing it, radial slats were cut by the band saw with the dimensions 400 × 70 × 7 mm and then dried to a moisture content of 8%. According to the test design used (Table 1), three-layered glulams (with two replicates per treatment) using an adhesive (with the concentration 50% and dry substance consumption of 150 g/m^2^) put on two lower and upper surfaces of the middle board were pressed (at 15 kg/cm^2^) inside an experimental hydraulic press with the press plate temperature of 160 °C for 20 min. After keeping the boards in the laboratory conditions for 2 weeks, they were trimmed by a circular saw to prepare the bending test specimens. The direction of pressure during the bending test was perpendicular to the glue line surface. The bending test was performed at the loading speed 5 mm/min using a universal tester (load cell-2 ton: Sanaf Co. LTD., Tehran, Iran) according to the EN 302-1 standard [27].

### 2.4. Characterization Analysis

The chemical compound of natural starch, modified starch, pure UF resin, and the mixture of the UF resin with the modified starch treated in alkali medium with the concentration 10% and 20% (after complete curing at the temperature 160) (as the index adhesives) were analyzed by FT-IR (Fourier transform infrared) spectroscopy using the pelletized samples. For this purpose, after mixing 100 mg potassium bromide (KBr) with 2 mg adhesive sample that was chopped into flour, the prepared samples were scanned using a Thermo Scientific Nicolet 6700 FT-IR Spectrometer (Thermo Fisher Scientific, Waltham, MA, USA) in the wave number range 600–4000 cm^−1^. The X-ray diffraction (XRD) patterns of the samples were recorded by a STOE-STADV (Germany) wide angle X-ray diffractometer using a CuKα radiation source with a wavelength of λ = 0.154 nm. 

### 2.5. Artificial Neural Network (ANN) as a Prediction Tool

The widest type of ANN used for prediction is the multi-layer perceptron (MLP). The typical example of the MLP structure is shown in Figure 1. The MLP Equation (1) is the mathematical statement of MLPs output as Figure 1.
(1)$$Y=g\left(\theta +\overset{m}{\underset{j=1}{\sum }}\vartheta_{j}\left[\overset{n}{\underset{i=1}{\sum }}f\left(w_{ij}X_{i}+\beta_{j}\right)\right]\right)\;$$
where Y is the prediction value of the dependent variable, *X_i_* is the *i*-th input value of the independent variable, *w_ij_* is the weight of connection between the *i*-th input neuron and the *j*-th hidden neuron, *β_j_* is the bias of the *j*-th hidden neuron, *ν_j_* is the weight of connection between the j-th hidden neuron and the output neuron, *θ* is the bias of the output neuron, and g(.) and f(.) are the active functions of the output and hidden neurons, respectively. 

In order to avoid underfitting and statistical bias on the one hand, and overfitting, high variance, and increasing time for training the network, the correct number of neurons in the hidden layer needs to be determined. For this porpoise, the best neuron numbers in the hidden layer can be calculated by different methods, such as the following: the number of hidden neurons should be less than twice the size of the input layer; the number of hidden neurons should be 2/3 the size of the input layer, plus the size of the output layer. According to obtained statistics, there was no difference between the mentioned methods and the trial-and-error process, which is described as the best way to obtain the hidden layer [21]. 

Training a MLP using a backpropagation learning algorithm as the commonest neural network algorithm to model various engineering applications [28] means to determine the best weight of the relations between neurons to achieve a minimum difference between the measured and estimated values of the dependent variable [29].

Training was performed by attempts made to receive different ANN models with different network configurations and by training the parameters. The models were tested using the test data set for the training process to test the network’s performance. For this purpose, three algorithms consisting of the Levenberg–Marquardt algorithm (trainlm), scaled conjugate gradient (trainscg), and Bayesian regularization (trainbr) were selected as feed-forward neural network models for training data sets. As a result, the ANN model producing the closest values to the measured values of MOR was chosen as the prediction model. This was performed based on some statistical criteria, such as R^2^, RMSE and MAPE showing the best performance of each model, so that the three-layer ANN architecture, including one input layer, one hidden layer, and one output layer was chosen (Figure 1). 

The formaldehyde to urea molar ratio (MR), the weight ratio of starch to urea added in second stage (WR), and the NaOCl concentration treating starch (AC) were chosen as the network input in the chemical treatment of the starch used while the variable MOR was used as the network output in the prediction model. Thus, there are three input neurons and one output neuron in the ANN model developed to predict the MOR. The best performance of the ANN model was obtained by statistical indicators showing the performance, such as the mean absolute percentage error (MAPE, Equation (2)), the root mean square error (RMSE, Equation (3)) and the determination coefficient (R^2^, Equation (4)) for a configuration with six neurons according to the Figure 1. Hence, the neuron configuration 3-6-1 was determined as the optimum configuration.
(2)$$RMSE=\sqrt{\frac{1}{n}\overset{n}{\underset{i=1}{\sum }}{\left(t_{i}-td_{i}\right)}^{2}}\;$$
(3)$$MAPE=\frac{1}{n}\overset{n}{\underset{i=1}{\sum }}\left(\frac{\left|t_{i}-td_{i}\right|}{t_{i}}\right)\;100\;$$
(4)$$R^{2}=1-\frac{\sum_{i=1}^{n}{\left(t_{i}-td_{i}\right)}^{2}}{\sum_{i=1}^{n}{\left(t_{i}-\bar{t}\right)}^{2}}\;$$
where *t_i_* is the measured value, *td_i_* is the predicted value, n is the total number of data, and $\bar{t}\;$ is the predicted mean value.

In the designed model, the tansig transfer function (tansig) in the hidden layer and the pure line transfer function (purelin) in the output layer were used as the activation functions. The Levenberg–Marquardt back-propagation algorithm was preferred as the training algorithm. The data in the range from −1 to 1 were normalized. Then, the normalized data were changed into the original values using an inverse normalization. The ANN programs were performed using MATLAB software, version R2015a, The MathWorks, Inc, Natick, MA, USA.

## 3. Results and Discussion

### 3.1. FTIR and XRD Characterization Analysis

FTIR and XRD analyses were conducted to investigate the possible interactions between oxidized starch and UF resin. FTIR spectra of native starch, modified starch, and synthesized adhesives are presented in Figure 2.

The presence of a band at 3200–3300 cm^−1^ in all samples is attributed to OH stretching vibrations. For native starch, the characteristic bands were mainly observed at 2922 cm^−1^, 1650 cm^−1^, 1364 cm^−1^, 1149 cm^−1^, and 1075 cm^−1^. The bands at 2922 cm^−1^ and 1364 cm^−1^ originated from C-H stretching vibrations of methylene (−CH_2_) groups [30]. The band at 1650 cm^−1^ is assigned to C-O bending vibrations associated with hydroxyl groups. The stretching vibrations of asymmetric C-O-C and C-O functional groups would cause the absorption bands at 1149 cm^−1^ and 1075 cm^−1^, respectively [31,32]. Based on previous reports, the carboxyl and aldehyde groups for oxidized starch appeared in the 1650–1735 cm^−1^ range [33]. For oxidized starch (b and c spectra), the band at 1650 cm^−1^ became stronger and it was also shifted to a lower wavenumber, confirming the formation of more aldehyde groups during the oxidation process. The symmetric stretching band of carboxyl groups (−COO) arising from oxidized starch might overlap with the band belonging to C-O bending vibrations of native starch, which resulted in band broadening. The band at ~3260 cm^−1^ was strengthened and widened, which was assigned to the formation of free hydroxyl groups as a result of chain scission in the oxidative process. The band’s broadening at 2922 cm^−1^ after oxidation is related to chain scission, which further evidenced the successful oxidation of starch [34]. For the UF/MS adhesive (c and d spectra), the appearance of two bands at 1470 cm^−1^ and 1150 cm^−1^ are attributed to the stretching vibration of the N-H group and the deformation vibration of the C-H group, respectively. This finding verified the transformation of oxidized starch and UF into adhesive resin via a polycondensation reaction [35]. Furthermore, the bands at ~3300 cm^−1^ and 2920 cm^−1^ belonging to O-H and C-H stretching vibrations were weakened, which confirmed the generation of polycondensation and partial intermolecular hydrogen bonds [34]. It was reported that N–H stretching of the primary amide in the UF resin overlapped with –OH groups of MS at 3300 cm^−1^. Indeed, the intra-molecular O–H of oxidized starch at 1640–1650 cm^−1^ overlapped with the C = O bond of the amide group in the UF resins [36].

An XRD analysis was performed to investigate the phase structure of the samples during the synthesis and curing, too. The diffractograms of native starch and modified starch are shown in Figure 3A. The XRD pattern of native starch showed prominent diffraction peaks at 2θ values of 15.31°, 17.5°, 19.43°, and 23.13°, which were the characteristic peaks of type A starch [37]. Modified starch oxidized by 10% and 20% NaOCl (MS10% and MS20%) exhibited similar diffraction peaks illustrating that the oxidation mainly occurred in the amorphous region of starch [38]. These findings are consistent with the previously published data [39,40,41,42]. Figure 3B presented the XDR diffractograms of the MS/UF adhesives synthesized by different F:U mole ratios. As could be seen, all of the characteristic peaks of oxidized starch disappeared or were significantly weakened, which revealed the alteration of the crystalline region into an amorphous structure. This verified the conversion of MS and UF into adhesive [42].

### 3.2. Experimental Results of Glulam’s MOR

The effect of MR, WR, and AC was examined on the bending strength and was modeled by ANN. Table 2 presents the bending strength values obtained as the result of experimental study together with the estimated values. In addition, the statistical evaluation of the effects of the independent variables was performed on the response being studied by ANOVA (Table 3). This statistical method can determine the difference or similarity between two or more data groups based on the comparison of the average value of the properties being examined. Based on the ANOVA results, the quadratic model was chosen. The effects of the linear parameters WR (x2), AC (x3), interactive parameters MR×WR (x1x2), MR×AC (x1x3), WR×AC (x2x3), and quadratic parameters MR (x1^2^), WR (x2^2^), and AC (x3^2^) on the bending strength were significantly and statistically different with *p* < 0.05%. Moreover, the non-significance of the lack of fit also shows the fitness of the quadratic model.

Based on Table 3, the F-value of the quadratic effect of AC (x3^2^) had the highest effect on the response with the highest value (1110) and the quadratic effect of MR (x1^2^), the interaction effects of MR×WR (x1x2) and WR×AC (x2x3) had the lowest effect on the response with the lowest value. However, when applying an average level of MR, the low level WR, and an average level AC, the maximum MOR (140 MPa) was obtained. 

### 3.3. Predicting MOR by ANN 

The predicted values of the experimental MOR and their error percentage in Table 2 and their analysis indicate that the predicted values resulting from ANN have a very low error percentage. This level of error is satisfactory for MOR so that it can be said that the ANN models have had a suitable performance to predict the bending strength of glulam. The maximum error percentage of the model does not exceed 1.55.

Three different models with a higher precision and three inputs were chosen to estimate the MOR so that the MR, WR, and AC were used as three inputs in the first layer and the MOR was used as the output in the last layer. The feed-forward neural network was trained by the Levenberg–Marquardt algorithm (trainlm), scaled conjugate gradient (trainscg), and Bayesian regularization (trainbr) algorithms. Meanwhile, the backpropagation learning algorithm was used in the feed-forward neural network with one hidden layer. In the produced model, the tansig activation and purelin functions were used between the input-hidden layer and hidden-output layer, respectively. The statistics R^2^, RMSE, and MAPE were used to evaluate the performance of each ANN model with three different training algorithms for MOR (Table 4). The statistical performances of the models were somewhat close to each other for training, testing, and validation data sets. Among the ANN models developed to estimate the MOR, the model MOR3 (3-6-1) with the highest R^2^ and lowest RMSE and MAPE with the BP training algorithm and 50 iterations offered the highest performance to estimate the response. Hence, the optimal network structure established based on R^2^ and RMSE criteria was offered in a network with six neurons in the hidden layer. 

Due to the high importance of MAPE as a statistic showing the performance of a model to make decisions, it had low values for predicting the MOR for the training, testing, and validation data sets (0.55, 0.903%, and 0.74%, respectively). The results show that the ANN approach has enough precision to predict the MOR. Hence, according to Table 3, it can be said that the proposed ANN model can predict MOR with a high precision when it is related to the real values. 

When evaluating the validity and precision of a neural network, the regression analysis often connects the actual and predicted values (Figure 4) for MOR in testing, validation, training, and all data sets. It should be mentioned that, if the determination coefficient approaches 1, the prediction precision increases [43]. It emphasizes that there is an excellent fit between the actual and estimated values. As it is observed in Figure 4, R^2^ values to predict the MOR for testing, validation, training, and all data sets are 0.9639, 0.9817, 0.9937, and 0.9903, respectively. R^2^ indicates that the obtained network describes at least 96% of the actual data of MOR. These values confirm the applicability of ANNs to predict glulam’s MOR.

Limited studies are conducted on the prediction of the bending strength of laminated wood products connected by bio-based adhesives. However, several studies are conducted on some strength properties of wood and wood-based products. The results obtained on the prediction of the strength behavior of these products based on R^2^ have been diverse. However, in almost all cases, it is confirmed that different ANN prediction models could offer an acceptable estimate. Tiryaki and Hamzacebi (2014) showed that using ANN, the strength of the thermally treated wood can be predicted with an R^2^ of more than 0.99 [29]. You et al. (2022) obtained a high R^2^ (0.98) when applying ANN to predict the mechanical properties of bamboo-wood composite [44]. Nguyen et al. (2019) obtained a very suitable estimate of the color change of the thermally treated wood in artificial weathering by ANN with the R^2^ = 0.92 [45]. 

The comparison of the predicted and actual values of the test and all data sets for MOR is presented in Figure 5, graphically. It is observed that the ANN outputs are very close to the actual outputs of MOR. This fact increases the applicability of the designed prediction model. These findings can also show that well-trained ANN models can be used effectively to predict glulam’s MOR in order that the number of tests or the test cost and time decrease.

An ideal ANN model can identify and apply complex nonlinear relationships between the inputs and outputs of any process with a high trainability [46]. Hence, it can be said that it is very important to calculate the relationships between the variables of the laminated products production and the properties being studied as the likely responses through ANN, i.e., produce products with a suitable quality with a very low energy consumption. In addition, the well-trained models can produce all intermediate values for optimization examinations [47]. This aspect of ANNs has shown that this method can be used successfully to optimize the production process conditions. As a result, the average values obtained can be used to optimize the preferred ANN model and achieve an optimal combination of the production parameters of a high quality product so that its result can be generalized, not only to laboratory, but also to industrial production and, in practice, some UF resin can be replaced by bio-based adhesives, according to the variables being examined. When replacing some UF resin with starch, Nazerian et al. (2022) showed that the ANN application could evaluate the glue line bonding strength in glulam and recommended the application of the modified starch [48]. Optimization of the application of the modified plant protein together with the MUF resin in the production of the polyurethane-core-based sandwich panels indicated that the application of the ANN optimization methods could offer an effective estimate of the mechanical properties of bio-based composites [49]. The comparison of different optimization and modeling methods showed that these methods can estimate the mechanical properties of the laminated products connected by the modified starch, even at high values of UF resin replacement [50].

Figure 6 shows the interactive effect of the factors being studied on the MOR at any of the three levels of the independent variable x1 (MR), x2 (WR), and x3 (AC). At the same time, it shows the deviation of the estimated values from the actual values for all three interactive effects x1x2, x1x3, and x2x3. It is observed that there is a rather perfect consistency and overlap between the estimated and experimental values. To show it, the error percentage of the estimated and experimental values is given in Figure 7. It can be seen that the error percentage ranges from 1.06 to 1.4%, which is very low, showing the perfect agreement between the estimated and experimental values.

The interactive effects of the independent variables on the MOR are given as a 3D plot in Figure 8a–c. It is evident from Figure 8a, where AC is at the middle level (15%), that the suitable level of MR is partly above 2:1 and the suitable level of WR is 10%. There is no positive effect on MOR, as the starch content increases even as the F to U molar ratio increases or decreases. Based on the ANOVA table, it is also observed, due to the non-significance of the direct effect x1 (MR) and the low F-value of the interactive effect of x1x2, that the effect of MR is low, according to the slope variations in the 3D plot at all levels in the interaction with WR.

The interactive effect of MR and AC on MOR is given in Figure 8b. As the alkali concentration increases by a little less than 20% when the F to U molar ratio is minimum, the MOR value is maximum, and as the F to U molar ratio increases, MOR decreases gradually. Based on the ANOVA table (Table 3), the F-value (6.34) indicating the effect of each variable shows a low effect, though significant, compared to other variables. Therefore, the intensity of the changes in MOR is affected more slowly interactively due to the change in each or both variables.

The interactive effect of WR and AC on MOR is shown in Figure 8c. When the bending strength is maximum and MR is at the middle level (2:1), WR (x2) is minimum, and AC (x3) is at the middle level. As WR becomes minimum, a continuity is observed in the MOR decrease. However, as AC decreases or increases beyond the middle level (15%), MOR decreases. This decrease is maximum when the alkali concentration is minimum. Based on the F-value in the ANOVA table, it is observed that the effect of this interaction is minimum (5.32) among all affecting variables.

Based on the ANOVA table, the F-value of every variable indicates that the direct effect of the factors has been high on the MOR. In Figure 9, the direct effect of all three independent variables on MOR and the fluctuations of the output estimated by ANN compared to the experimental value are shown. It is observed that the estimated values of MOR resulting from the effect of the independent variables overlap with the experimental values at all used levels of every variable, and it is also confirmed by the statistics R^2^ and RMSE. At the same time and under the effect of the variables, MOR was affected by the variables differently. As MR increases from 1.5 to 2, MOR increases. However, as it increases more and reaches 2.5, MOR decreases. As the UF resin to MS weight ratio increases to 80%, the increase in MOR is slow. However, applying the minimum starch (10%), the intensity of the increase in MOR has increased and becomes maximum. It is observed that, as the treatment concentration increases from 10% to 15%, the strength increases, and when it increases beyond it to 20%, the bending strength decreases again. 

In the bending test, due to the difference in the deformation direction of the layers under pressure compared to the layers under tension, the shear stress can result in the failure of the samples. It was observed that the samples with the highest or lowest formaldehyde and also with the highest starch to UF weight ratio have a failure at the larger area in the intraphase or interphase region of the adhesive due to the dominance of the shear stress over the bending stress. However, as the F to U molar ratio approaches the middle level (2) and as the ratio of starch approaches 20%, there is a failure under the tensile stress so that it begins from the wood and moves along the wood layer fibers. Starch oxidation can decrease the adhesive viscosity due to the failure of the inter- and intra-molecular connections [51] and result in the excessive penetration of the adhesive into the internal layers of the wood that prevents the formation of a uniform film of resin on the bonding line and results in stress concentration regions. Despite the effect of oxidation on granules, the structure of which has changed and have decreased the starch viscosity, the system’s viscosity is still more than the pure resin. Furthermore, during the treatment, the structure of many granules changes depending on the oxidizing agent and the treatment temperature, while many other granules remain unchanged. Due to the gelatinization viscosity of the unchanged starch granules, an evident increase occurs in the viscosity of the adhesive system (especially when applying less oxidation through using less concentrated alkali) [52] and wood penetrability decreases, and a cohesive failure occurs in the loading. Due to the application of the alkaline oxidation treatment to modify the natural starch and acidic treatment to produce the starch adhesive, the polymer chain containing new and more bulky groups is exposed to make connections with the extra formaldehyde in the UF resin so that a cross-linked network structure is formed due to it and the adhesive viscosity also increases. In other words, while starch is treated in the oxidation process to produce the adhesive, the molecular weight of the starch decreases effectively, and the gelatinization viscosity is decreased effectively [53]. 

As the solid substance and viscosity increase on average, the solid structures of the modified adhesive tend to get bigger through polycondensation and hence, the molecular weight of the resin system increases. Meanwhile, due to the formation of more hydroxymethyl in the modified starch, the formation of a hydrogen bond is more possible with the UF resin. However, the curing time decreases, which is the criterion of resin activity. As the modified starch increases more, or a more intense treatment of starch is applied, the effect becomes inverse and the curing and complete copolymerization in the modified starch are limited due to the thermoplastic nature of starch [8]. The chemical treatment of starch promotes grafting in the starch chain due to the weak mobility of starch macromolecules [54]. During this process, the molecular weight of starch decreases and active sites will be more exposed. Hence, the formation of a starch macromolecular colloid becomes more likely with resin molecules [55]. The copolymer content also increases as the grafting parameter increases, which consequently improves the compatibility between starch and the polymer [55]. As a result, the bonding strength improves which improves the resistance to the horizontal shear stress created during the bending of the layers. However, the results show that, as the treatment intensity increases, the grafting parameter decreases due to homopolymerization that has overcome the graft-copolymerization [56]. As the oxidation intensity increases, the active structure of starch is destroyed and hence, the grafting parameter decreases [55] and the strength decreases. 

Generally, the starch molecules in gel and paste forms are related to retrogradation. The chemical treatment of starch decreases the molecular weight of starch and the steric hinderance, providing a good opportunity for grafting between starch molecules and resin. Hence, the quality of adhesives increases when the gelatinization of starch molecules is hindered in the system [55]. However, when the treatment conditions become harder, the active structure of starch is destroyed. The starch chain becomes smaller and is gelated easily. Hence, the treatment prevents retrogradation to some extent.

In addition, based on the SEM images, it is evident that, during the oxidation, the spherical shape of starch can disappear completely and the surface structure becomes rough or bar-shaped when it is added to urea and the OS-U adhesive is formed, showing the polycondensation reaction between urea and the modified starch [42]. However, at the same time when using higher contents of the oxidized starch, especially together with a stronger alkaline treatment, due to much more frequent failures of starch polymers and the opening of hemiacetal loops of glucose monomers of amylose and amylopectin chains of starch at the same time, viscosity declines strongly again so that it can be less than the resin viscosity, which is less than that of starch. However, when the oxidized starch treated with a 15% alkali concentration is applied at the middle level, the adhesive has a higher viscosity, and the more uniform distribution of the adhesive improves on the layer. Moreover, based on the SEM micrograph analysis, it became clear that the application of the starch adhesive can result in a more uniform distribution of the adhesive on the substrate so that a more identical compressed structure is created inside the board [7]. Consequently, under the effect of these two factors on one hand and the compatibility between the UF resin and starch on the other hand, a better connection is made between the substrate and the adhesive.

When applying the modified starch, carbonyl and carboxyl groups react with amino groups (−NH_2_) of resin in the presence of a catalyst, such as ammonium chloride, that develops a stronger cohesion together with the branched structure of amylopectin so that it was observed that the failure of the samples containing a suitable level of starch occurred mainly in the wood layers, not on the glue line. In addition, the chemical reaction between the active aldehyde and carboxyl groups of the starch molecules and the hydroxyl groups in wood can improve the bonding strength more, which is very effective for bearing the horizontal shear force in the bending test and the stress can be transferred from the surface under pressure to the surface under tension. It seems that under optimal ideal conditions, where the adhesive molecule is limited to the surface layers to some extent and the gel nail can form after hardening and more penetration of resin is prevented, rupture occurs completely in the wood region. 

It is proved that the strength properties of wood composites decrease as the crystallinity index of resins increases. This index decreases using the UF resins known as a crystalline substance during the curing [7] because the UF resin structure changes from microcrystalline to amorphous when it is cured in wood [57]. Due to the significant decrease in the crystallinity index of the oxidized starch adhesive when applying the UF resin [7], it can be expected that the adhesive strength increases at lower levels of the used starch adhesive. As a criterion to describe the quality of the chemical bond in wood products, the differential scanning calorimetry (DSC) analysis has proved that the glass-transition temperature decreases due to adding starch to the UF resin because more UF in the adhesive has increased the melting point of the adhesive due to more cross-linkages being created by UF compared to the modified starch [10]. In addition, when applying a stronger treatment of oxidation, instead of the formation of (−CH_2_−O−CH_2_−) ether bridges that are more likely to occur in the more moderate conditions of oxidation treatment, (−CH_2_−) methylene bridges form at high pressure temperature [58] that can increase the bonding strength. However, it seems that it is not unlimited, and the effect of oxidation is that the stronger treatment is reversed due to the destruction of the glycoside bond and opening of hemiacetal loop of starch.

### 3.4. GA-ANN Optimization

The problem of the optimization of the strength properties of wood composites must be defined as mathematical model equations or the fitness function. These equations are given as the dependent functions of measured value and construction parameters. Due to the complexity of the models, more accurate mathematical models include linear and nonlinear components. However, in practice, second-order polynomials are enough to develop a mathematical model to describe the production and construction process. For this purpose, based on the independent variables x1, x2, and x3, the multi-objective functions and nonlinear constraint functions were used respectively to determine the actual effect (in a definite interval) and estimated effect (in indefinite interval) of each independent variable as a direct, interactive, and quadratic form on each source, including f(x1,x2), f(x1x3), and f(x2x3) using the genetic algorithm (Table 5). The minimization of the objective function value of equations illustrated in Table 5 was exposed to the used limits using the nonlinear constraint function. Meanwhile, the optimal limits of using the independent variables used were 1.6215:1 for MR, 17.67% for WR, and 10.6% for AC, which are the best combination of parameters used, leading the minimum values of the objective functions to the nonlinear constraint functions.

## 4. Conclusions

The present study has predicted the effect of the application of the starch modified at different concentrations of NaOCl in the UF resin synthesis with different F to U molar ratios as the inputs on the bending strength of glulam as the outputs using the ANN and has optimized it using the ANN-GA. The results showed that: UF resins were successfully modified by adding OS prepared by NaOCl oxidation. FTIR and XRD analyses detected both aldehyde and carboxyl groups in the OS. The OS was cross-linked with UF resin by forming the ester groups.The ANN-GA model with three neurons in the input layer and six neurons in a hidden layer showed the best performance of optimization with R^2^ = 0.9937, RMSE = 0.72, and MAPE = 0.55.The proposed optimal network predicted a MOR value of more than 126 MPa, validated by experiments.Based on the optimization equations of the genetic algorithm, the difference between the optimal values given by the multi-objective functions based on the actual response and the values given by the nonlinear constraint function based on the estimated response (outputs) was minimum.According to the statistics and the coefficients of the optimization equations, the concentration of the chemical solution treating the starch has been the main factor affecting the changes in the response level.

## Figures and Tables

**Figure 1 materials-15-09074-f001:**
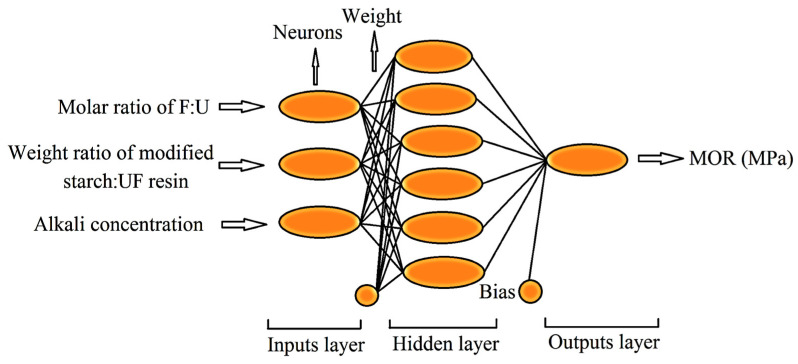
The ANN architecture used as the prediction model of MOR.

**Figure 2 materials-15-09074-f002:**
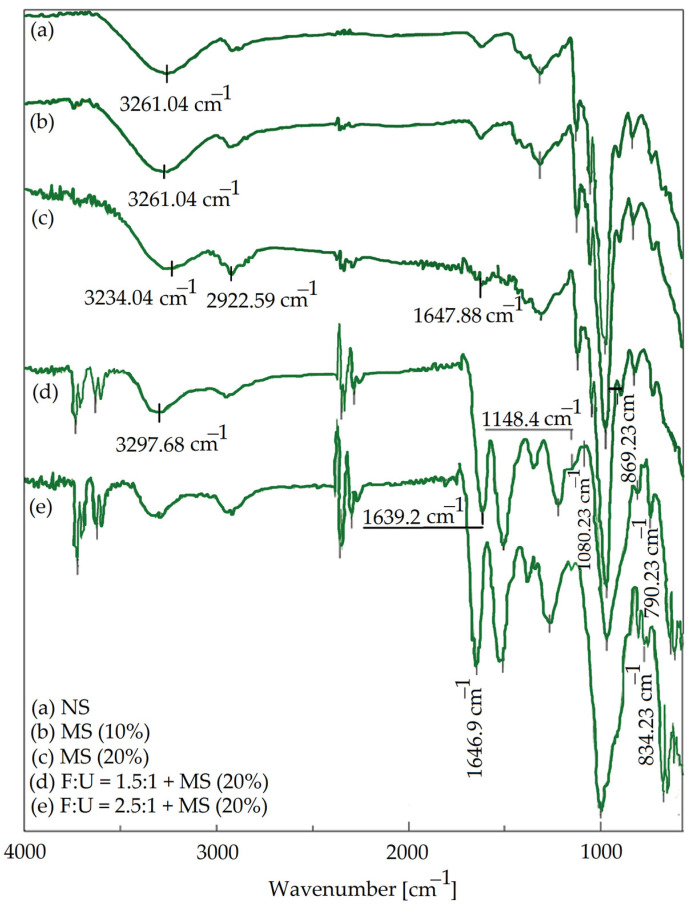
FTIR spectra of neat starch (**a**); modified starch by 10% (**b**) and 20% (**c**) NaOCl and modified starch/urea/formaldehyde adhesives synthesized by different F:U molar ratios (**d**) 1.5:1 and (**e**) 2.5:1.

**Figure 3 materials-15-09074-f003:**
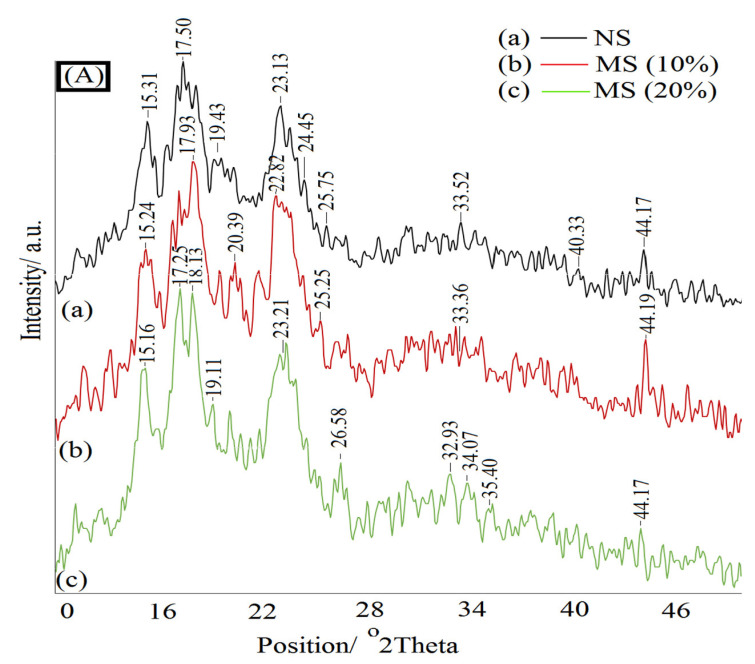
XRD diffractograms of (**A**) neat starch and modified starch by 10% and 20% NaOCl; (**B**) modified starch/urea/formaldehyde adhesives synthesized by different F:U molar ratios.

**Figure 4 materials-15-09074-f004:**
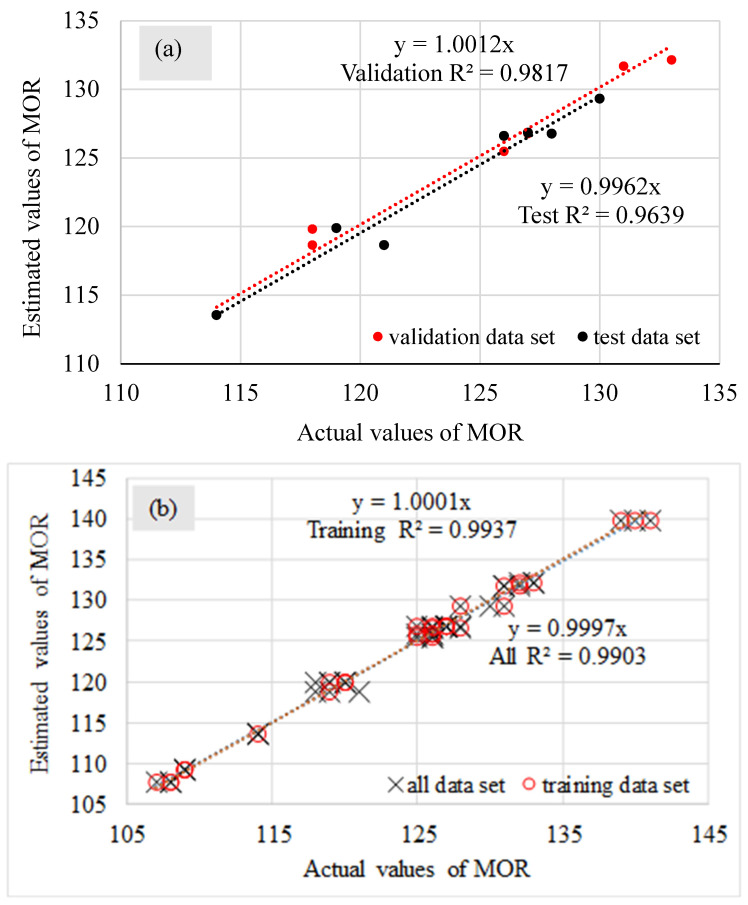
The regression curve of validation and test (**a**) and training and all (**b**) data sets.

**Figure 5 materials-15-09074-f005:**
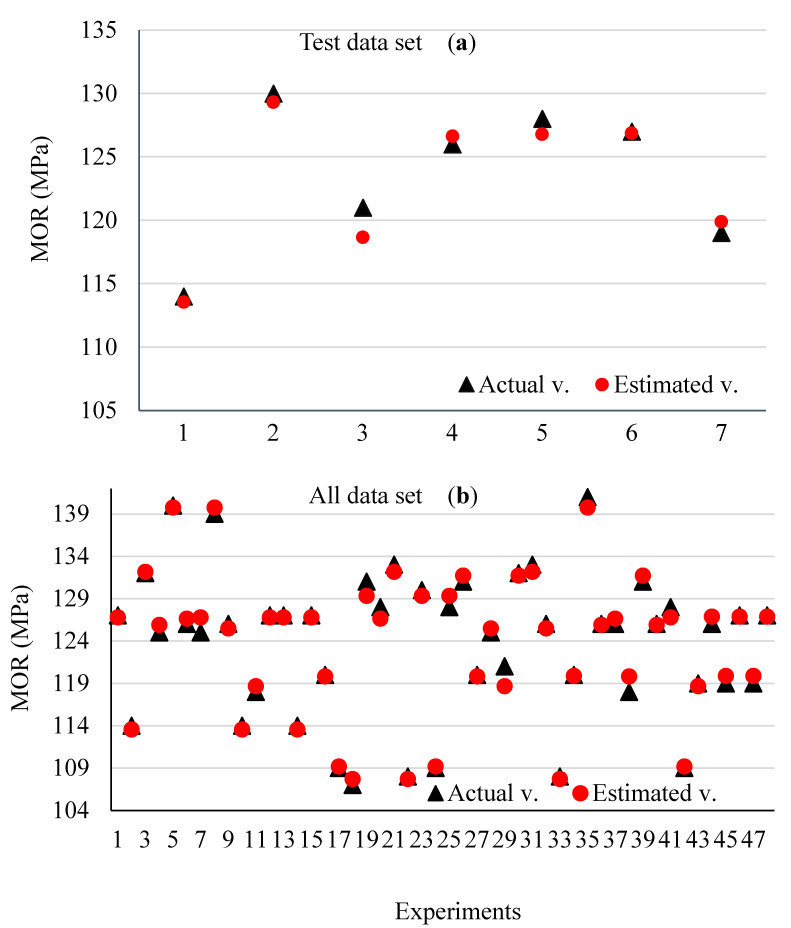
The comparison of the real and predicted values of the test (**a**) and all data (**b**) sets.

**Figure 6 materials-15-09074-f006:**
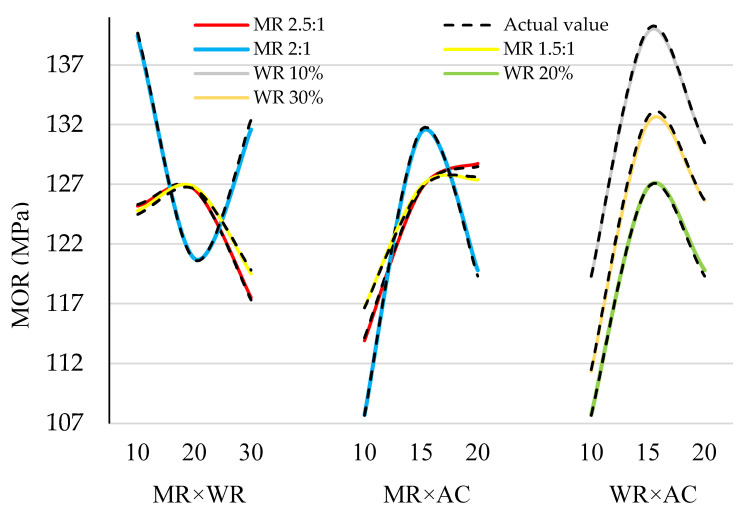
The comparison of the interactive real and estimated effects of (**left**) ANN MR × WR, (**center**) MR × AC, and (**right**) WR × AC on the bending strength.

**Figure 7 materials-15-09074-f007:**
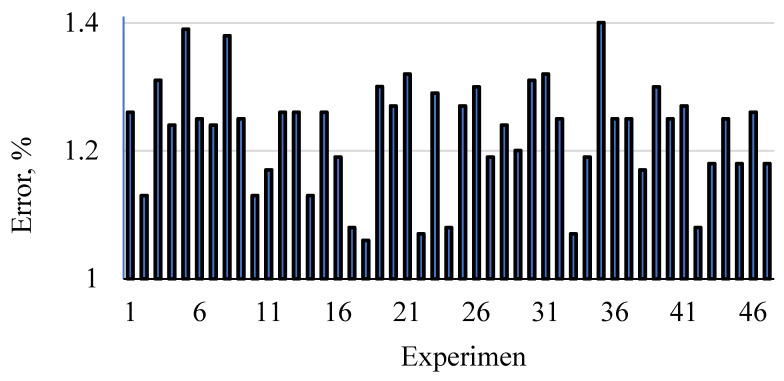
Error of bending strength estimation when applying the ANN modeling.

**Figure 8 materials-15-09074-f008:**
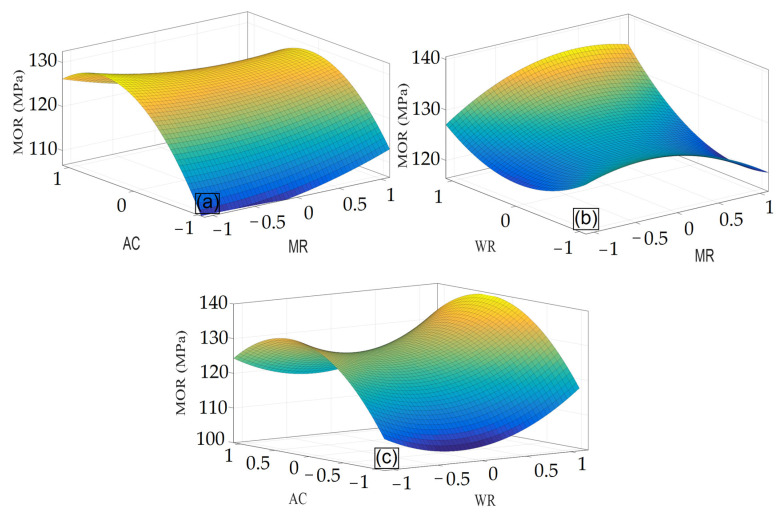
The interactive effect of (**a**) MR×WR, (**b**) MR×AC, and (**c**) WR×AC on the bending strength.

**Figure 9 materials-15-09074-f009:**
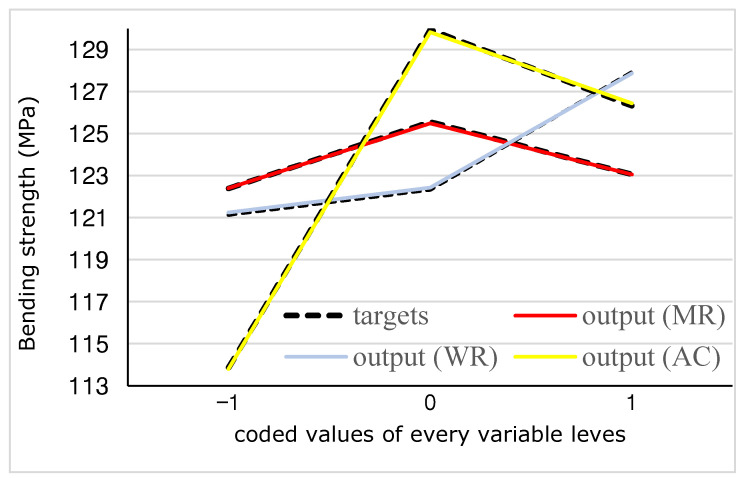
The direct real and estimated effect of the variables on the bending strength.

**Table 1 materials-15-09074-t001:** Experimental design.

Run	MR	WR	AC	Run	MR	WR	AC
1	2	80	15	25	2.5	90	20
2	2.5	70	10	26	1.5	90	20
3	2	70	15	27	2	80	20
4	1.5	70	20	28	2.5	70	20
5	2	90	15	29	1.5	90	10
6	1.5	80	15	30	1.5	90	20
7	2	80	15	31	2	70	15
8	2	90	15	32	2.5	70	20
9	2.5	70	20	33	2	80	10
10	2.5	70	10	34	2.5	90	10
11	1.5	90	10	35	2	90	15
12	2	80	15	36	1.5	70	20
13	2	80	15	37	1.5	80	15
14	2.5	70	10	38	2	80	20
15	2	80	15	39	1.5	90	20
16	2	80	20	40	1.5	70	20
17	1.5	70	10	41	2	80	15
18	2	80	10	42	1.5	70	10
19	2.5	90	20	43	1.5	90	10
20	1.5	80	15	44	2.5	80	15
21	2	70	15	45	2.5	90	10
22	2	80	10	46	2.5	80	15
23	2.5	90	20	47	2.5	90	10
24	1.5	70	10	48	2.5	80	15

**Table 2 materials-15-09074-t002:** Real and estimated values and error percentage at any treatment/repeat.

Run	Actual Value (MPa)	Estimated Value (MPa)	Error (%)	Run	Actual Value (MPa)	Estimated Value (MPa)	Error (%)
1	127	126.7825	1.2600	25	128	129.3212	1.2699
2	114	113.5608	1.1300	26	131	131.7027	1.2999
3	132	132.1597	1.3099	27	120	119.8225	1.1900
4	125	125.8852	1.2399	28	125	125.4881	1.2399
5	140	139.7595	1.3900	29	121	118.6683	1.2002
6	126	126.6249	1.2499	30	132	131.7027	1.3100
7	125	126.7825	1.2398	31	133	132.1597	1.3200
8	139	139.7595	1.3799	32	126	125.4882	1.2500
9	126	125.4881	1.2500	33	108	107.6981	1.0700
10	114	113.5608	1.1300	34	120	119.8915	1.1900
11	118	118.6682	1.1699	35	141	139.7595	1.4000
12	127	126.7825	1.2600	36	126	125.8851	1.2500
13	127	126.7825	1.26001	37	126	126.6249	1.2499
14	114	113.5608	1.1300	38	118	119.8225	1.1698
15	127	126.7826	1.2600	39	131	131.7027	1.2999
16	120	119.8225	1.1900	40	126	125.8851	1.2500
17	109	109.1891	1.0799	41	128	126.7826	1.2700
18	107	107.6981	1.0599	42	109	109.1891	1.0799
19	131	129.3211	1.3001	43	119	118.6683	1.1800
20	128	126.6249	1.2701	44	126	126.8605	1.2499
21	133	132.1597	1.3200	45	119	119.8915	1.1799
22	108	107.6981	1.0700	46	127	126.8605	1.2600
23	130	129.3211	1.2900	47	119	119.8915	1.1799
24	109	109.1891	−1.0799	48	127	126.8605	1.2600

**Table 3 materials-15-09074-t003:** ANOVA for response surface quadratic model.

Source	Sum of Squares	df	Mean Square	F-Value	*p*-Value	
Model	3390	8	424	311	<0.0001	***
WR (x2)	342	1	342	251	<0.0001	***
AC (x3)	1190	1	1190	875	<0.0001	***
MR×WR (x1x2)	8.63	1	8.63	6.34	0.0161	*
MR×AC (x1x3)	15.9	1	15.9	11.7	0.0015	**
WR×AC (x2x3)	7.25	1	7.25	5.32	0.0264	*
MR^2^ (x1^2^)	6.08	1	6.08	4.47	0.041	*
WR^2^ (x2^2^)	695	1	695	510	<0.0001	***
AC^2^ (x3^2^)	1510	1	1510	1110	<0.0001	***
Lack of Fit	23.4	6	3.91	4.35	0.244	NS

Note: * significant at 95%, ** significant at 99%, *** significant at 999%.

**Table 4 materials-15-09074-t004:** Comparison of ANN models for the estimation of MOR.

**Tra.**	**ANN**	**It.**	**Training**			**Testing**			**Validation**		
**Algorithm ***	**Stru.**		**R^2^**	**RMSE**	**MAPE**	**R^2^**	**RMSE**	**MAPE**	**R^2^**	**RMSE**	**MAPE**
trainlm trainscgtrainbr	3-6-1 3-6-1 3-6-1	50 50 50	0.996 0.965 0.994	0.627 2.23 0.72	0.54 1.51 0.55	0.565 0.856 0.964	11.6 3.83 1.11	8.15 2.57 0.903	0.989 0.955 0.982	1.45 2.51 0.88	1.22 2.23 0.74

* “trainlm”, “trainscg”, and “trainbr” are the Levenberg–Marquardt, scaled conjugate gradient, and Bayesian regularization algorithms, respectively.

**Table 5 materials-15-09074-t005:** The multi-objective and nonlinear constraints functions of the interactive effects of x1x2, x1x3, and x2x3 used to optimize the response being studied using the GA approach.

Source Normalized	Multi-Objective Function (Based on Actual Values)		Nonlinear Constraint Function (Based on Estimated Values)	
f(x1,x2) f(x1,x3) f(x2,x3)	124 + 0.3333x1 + 3.267x2 − 5x1^2^ − 0.8333x1x2 +4.667x2^2^ 129 + 0.3333x1 + 6.233x3 + 2265x1^2^ − 0.8333x1x3 −10.9x3^2^ 127 + 3.267x2 + 6.233x3 + 8.856x2^2^ − 0.75x2x3 − 13.98x3^2^		1.25 + 0.3084x1 + 3.458x2 − 3.789x1^2^ + 2.786x1x2 + 4.682x2^2^ 1.26 + 0.2001x1 + 6.2x3 + 1.602x1^2^ − 3.145x1x3 − 10.33x3^2^ 1.26 + 3.448x2 + 6.298x3 + 7.839x2^2^ − 0.5722x2x3 − 12.56x3^2^	
Optimal coded and actual values of inputs with response for every source (function)				
Source	Input (x1, MR)	Input (x2, WR)	Input (x3, AC)	Opt. response value
f(x1,x2) f(x1,x3) f(x2,x3)	−0.754 (1.6215:1)	−0.233 (17.67%)	−0.988 (10.6%)	120.249 112.806 106.727

## Data Availability

Not applicable.

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
