# Peer review of "An Improved Optimization Model to Predict the MOR of Glulam Prepared by UF-Oxidized Starch Adhesive: A Hybrid Artificial Neural Network-Modified Genetic Algorithm Optimization Approach"

_materials, 2022, doi:10.3390/ma15249074_

Round 1

Reviewer 1 Report

Recommendation: Major revision before reconsideration

Comments:

It is useful to obtain the certain MOR by the optimized model based on GA-ANN, and this is meaningful for industrial production. However, the innocation is not very clear in the paper and some necessary condition still needs to be addressed. 

Some comments and suggestions are listed as follows:

1.      In the Abstract, please give the full name of these abbreviations (RMSE and MAPE).

2.      For the introduction, the innovation is not very clear. Why authors want to use the ANN and genetic algorithm, and what benefits will be obtained from their combination?

3.      Line 184-186, how to implement the trial and error process? What is the rule? For the figure 1, why the hidden layer has six neurons?

4.      Line 193-198, what is the rule to choose the ANN model? and what is the best model? Please show the best model in detail. By the way, will the best model be used to prediction without any revision?

5.      Line 202, why only three neurons were chosen? Does these three neurons were the key factors for the MOR? How about the producing conditions (time, pressure, temperature) and raw material conditions (source, thickness, and arrangement).

6.      Line 203-204, last paragraph, author mentioned the prediction model was from training, and here gives that the best model was obtained by statistical indicators. Why? What is difference between those two models?

7.      Line 216, why this algorithm was chosen?

8.      For the section 3.1, what is the role of FTIR and XRD in this paper? Please clarify it in a proper place.

9.      For the Conclusions, the conclusion needs to be improved to show a concise and important results based on the above investigation. Moreover, can the optimized model predict the best MOR by giving a feasible adhesive formula? If so, what it is?

10.   Line 534, the abbreviation of GA-ANN was mentioned, what is the difference between it and ANN-GA. I guess this is probably a same thing.

Reviewer 2 Report

The manuscript presents an improved optimization model to predict the MOR of Glulam prepared by UF-oxidized starch adhesive. Besides, a hybrid artificial neural network-modified genetic algorithm optimization approach was used.

The Introduction is adequate, providing the necessary context and including relevant and up-to-date references. However, it is necessary to indicate more clearly the research gap, the novelty of the topic, and the expected impact of its results. The research design is adequate, including several experimental tests and prediction models. Moreover, the materials and methods are adequately described. The results are clearly presented and support the conclusions. In this reviewer's opinion, the manuscript has valuable scientific information to be published in the context of prediction models for glulam's MOR. Therefore, it is recommended its acceptance for publication after minor corrections. 

Specific comments:

Page 5, line 179: Please change "Wijis" to "wij is"

Page 8, line 283: The units of measurement of MOR (MPa) should be added in Table 2.

Table 2: Some runs are repeated (for example, 27, 28, and 29), and some are missing (for example, 25, 26, and 48). Please clarify.

Page 9, line 285: Please fix some super indexes (for example, x32).

Page 9, line 286: Please change (1110) to (1510).

Reviewer 3 Report

This study deals with the prediction of mechanical properties of glued laminated timber with UF-oxidized starch adhesive. For the material behaviour prediction an artificial neural network algorithm was used. In general, the content of this study is interesting. However, the research tasks/questions are not clearly pointed out. Considering from the title the optimization of the material property prediction algorithm is the most important part. However, this expression is changing during the reading of the manuscript. Also, chemical analysis and their results were presented much more as the reader could imaging by reading the title. In the conclusion section, the topic of material property prediction with ANN has again a prominent part. The authors should considered these aspects.  

Section 3.1: Please use superscript numbers in the units of the wave numbers (e.g. cm-1); 

Section 3.1: What do the models (e.g. x1x2, x1x3, ...) mean? At this stage the models were not defined. 

Table 2: Please add the units (e.g. MPa or N/mm²) of this measurement results; Is the actual value a mean of different measurements? Why do you have runs of number 27, 28 and 29 twice? Why are the runs with the number 24 and 25 missing? 

Table 4: What does the term “LM”, “SCG” and “BR” mean? 

Figure 5: The reader cannot understand these two charts. Why did you use a line diagrams? Can I say that the between the experiment 1 and 2 (e.g. 1.5 experiment) is a value of MOR?  

Figure 6: What do the terms (e.g. Es 2.5, Acv., …)? 

Figure 7: Why did you use line diagram? Maybe a histogram could be a better alternative compared to a line diagram? 

Table 5: Please check the results of your model. If they are correct then the present model is strongly influenced by a variable, which is not known or even analysed.

Round 2

Reviewer 3 Report

The authors reworked the manuscript following the reviewers' comments (e.g. Figures, model, content). However, at the beginning of the manuscript it is not clear what are the terms x1, x2 or x3, only in line 408 were the variables assigned to different measures/material characterizations. However, this assignment should have been made earlier, e.g. in line 308 or section 3.2, otherwise the reader cannot understand for example Table 3. If this change has been made, the manuscript can be recommended for publication. I do need to review the manuscript again.
